# Normalized Difference Vegetation Index Temporal Responses to Temperature and Precipitation in Arid Rangelands

**Ernesto Sanz** [1,2,*], **Antonio Saa-Requejo** [1,3], **Carlos H. Díaz-Ambrona** [1,4], **Margarita Ruiz-Ramos** [1,4], **Alfredo Rodríguez** [1,5], **Eva Iglesias** [1,6], **Paloma Esteve** [1,6], **Bárbara Soriano** [1,6] and **Ana M. Tarquis** [1,2]

1   CEIGRAM, Universidad Politécnica de Madrid, 28040 Madrid, Spain; antonio.saa@upm.es (A.S.-R.);
    carlosgregorio.hernandez@upm.es (C.H.D.-A.); margarita.ruiz.ramos@upm.es (M.R.-R.);
    alfredo.rodriguez@uclm.es (A.R.); eva.iglesias@upm.es (E.I.); paloma.esteve@upm.es (P.E.);
    barbara.soriano@upm.es (B.S.); anamaria.tarquis@upm.es (A.M.T.)
2   Grupo de Sistemas Complejos, Universidad Politécnica de Madrid, 28040 Madrid, Spain
3   Evaluación de Recursos Naturales, ETSI Agronómica, Alimentaria y Biosistemas,
    Universidad Politécnica de Madrid, 28040 Madrid, Spain
4   AgSystems, ETSI Agronómica, Alimentaria y Biosistemas, Universidad Politécnica de Madrid,
    28040 Madrid, Spain
5   Departamento de Análisis Económico y Finanzas, Universidad de Castilla-La Mancha, 45071 Toledo, Spain
6   Economía Agraria y Gestión de los Recursos Naturales, ETSI Agronómica Alimentaria y Biosistemas,
    Universidad Politécnica de Madrid, 28040 Madrid, Spain
*   Correspondence: ernesto.sanz@upm.es

**Abstract:** Rangeland degradation caused by increasing misuses remains a global concern. Rangelands have a remarkable spatiotemporal heterogeneity, making them suitable to be monitored with remote sensing. Among the remotely sensed vegetation indices, Normalized Difference Vegetation Index (NDVI) is most used in ecology and agriculture. In this paper, we research the relationship of NDVI with temperature, precipitation, and Aridity Index (AI) in four different arid rangeland areas in Spain's southeast. We focus on the interphase variability, studying time series from 2002 to 2019 with regression analysis and lagged correlation at two different spatial resolutions ($500 \times 500$ and $250 \times 250$ m$^2$) to understand NDVI response to meteorological variables. Intraseasonal phases were defined based on NDVI patterns. Strong correlation with temperature was reported in phases with high precipitations. The correlation between NDVI and meteorological series showed a time lag effect depending on the area, phase, and variable observed. Differences were found between the two resolutions, showing a stronger relationship with the finer one. Land uses and management affected the NDVI dynamics heavily strongly linked to temperature and water availability. The relationship between AI and NDVI clustered the areas in two groups. The intraphases variability is a crucial aspect of NDVI dynamics, particularly in arid regions.

**Keywords:** biometeorology; agrometeorological information; data analysis; grassland; MODIS

## 1. Introduction

Rangelands cover almost 33% of ice-free land globally. Monitoring rangeland is a key aspect of stopping its degradation. Severe degradation is causing the disappearance of 5–10 million hectares of agricultural lands every year [1,2]. Rangelands management is complex as rangelands are not spatially homogeneous; those with erodible soils and palatable vegetation are more prone to degrade than others. Arid and semiarid rangelands tend to have erodible soils and variable precipitation regimes, making them more sensitive to degradation [3,4]. In addition, different types of rangeland depend on land use, plant communities, soil, and climatic conditions. These include grazed wastelands such as stubble from mainly rainfed cereal crops, other croplands and fallow lands, grasslands, scrublands, and open woodlands [5]. Using remote sensing to monitor these areas and

improving the land's management practices can aid in preventing the degradation and reinforcing the sustainability of these ecosystems [2].

Remote sensing techniques are increasingly used by ecologists and agriculturalists [6–9], as a cost-effective means to measure the characteristics, biomass, and extent of vegetation [10]. Normalized Difference Vegetation Index (NDVI) is the most widely used vegetation index by rangeland ecologists [11]. It can be obtained at different spatial resolutions, the most common at $500 \times 500$ m$^2$ and $250 \times 250$ m$^2$, named Low and Medium Resolution (LR and MR). NDVI shows a good correlation with biomass of different vegetation types in arid and semiarid areas [12,13]. Another useful index is the NDVI anomaly ($Z_{NDVI}$). This index has shown robust results to identify damaged vegetation, and therefore it is especially suited to analyze the status of semiarid and arid areas [14].

Temperature and precipitation are mostly studied as drivers of NDVI [15–23]. However, the interactions among them and with other factors are still not fully comprehended [24]. Temperature effects on NDVI are significant when water is available in the ecosystem; in these circumstances, precipitation plays a minor role. However, this relationship grows more assertive in arid regions, where water availability seems to be one of the main drivers of NDVI [25–27]. Furthermore, some authors [28] suggest studying additional climatic factors further to understand thermal and hydric stress effects on NDVI patterns. Other climatic variables that have been reviewed are soil moisture, evapotranspiration, and land use cover [16,22,23]. The relationships of NDVI with these climatic variables differ by season and vegetation type and biome. Another factor that makes it more difficult to disentangle the interactions is the effect of human activities on both NDVI and the ecosystem itself, especially when it comes to overexploitation such as overgrazing or water overuse [29]. Therefore, it is essential to deeply understand NDVI and meteorological variables' relationship, especially accounting for the differences in this relationship between seasons and across types of land management. This is particularly relevant for agrometeorological indexes that should be calculated during the year, as it is in the case of rangelands [30].

NDVI and its relationship with meteorological variables have reported different results depending on the analyzed spatial scale. Tarnavsky et al. (2008) [31] in Arizona (USA) and Stefanov et al. (2005) [32] in different locations worldwide, described differences in spatial heterogeneity. Peng et al. (2017) [33] studied different spatial scales in China, which exhibited different correlations depending on coarser or finer scales; all were performed at monthly or annual temporal scale. On the other hand, human activities negatively correlated with NDVI at a Medium Resolution (MR) while it exerted a positive correlation at a lower resolution. Therefore, considering different spatial scales to identify the causes of NDVI patterns remains a challenging task.

The Aridity Index (AI) represents water availability, and different expressions have been developed [34]. We used a modified version of a widely accepted ratio of annual precipitation to the annual potential evapotranspiration in this work, developed by the United Nations Environment Programme in 1992 [35,36]. This index has been used to quantify droughts and estimate possible changes in climate regimes [37,38], and manage afforestation and reforestation projects and prioritize and assess future conservation efforts in rangelands [39,40]. It has also been previously compared to vegetation indexes [41].

The objective of the study was to evaluate the responses of the rangeland NDVI to temporal dynamics of temperature and precipitation for an arid environment. To evaluate it, NDVI series from four areas, corresponding to different types of rangeland in Murcia (southeast of Spain), were acquired for 2002 to 2019, at two spatial resolutions (LR and MR). The correlation between the NDVI and the climatic variables and the AI was used to gain better insight into the patterns of rangelands to monitor and characterize them and optimize the management accordingly.

## 2. Materials and Methods

### 2.1. Area of Study

The area selected is located in Murcia province, in the southeast of Spain. The area has a Mediterranean arid climate with annual precipitation less than 300 mm, although there are regional variations [42]. Four square areas with 2.5 km sides were selected for this study. They are situated in the vicinity of a meteorological station, covering three different agricultural regions of Murcia: Northeast, Segura River, and Northwest (Figure 1, Table A1 in Appendix A). The average temperature varies amongst the four areas from 14.7 to 17.3 °C, and the average accumulated precipitation per eight-day period ranges from 262 to 348 mm.

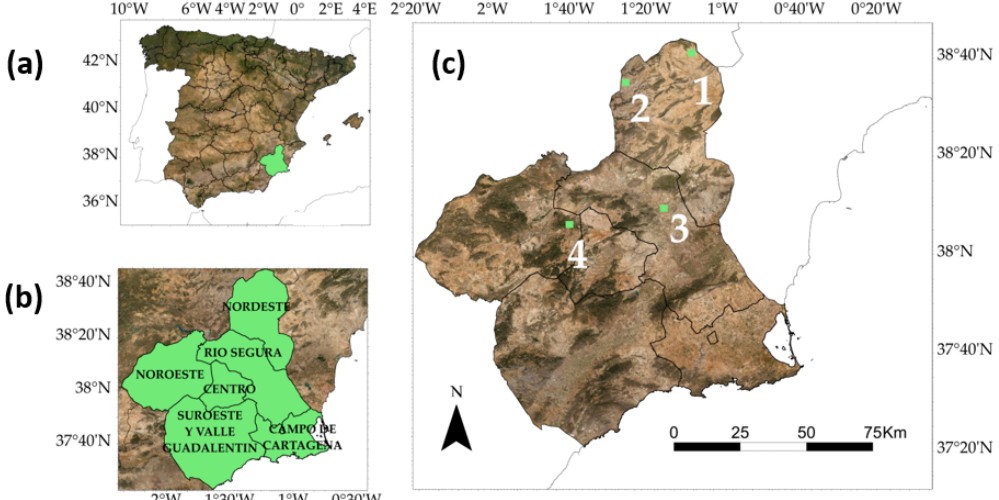

**Figure 1.** Location of the study area. (**a**) Autonomous Community of Murcia. (**b**) Agricultural regions of Murcia. (**c**) Selected areas in three agricultural regions of Murcia province. Numbers refer to the sampling areas. Source base map: Invierno 2020. Gobierno de España y Comunidad Autónoma de Murcia. CC-BY 4.0 scne.es 2020.

All the areas are used as rangeland, two predominantly herbaceous (A1 and A2) and two mainly covered by trees (A3 and A4). Area 1 (A1) is mostly covered by stubble from cereal crops. Area 2 (A2) is almost entirely covered by mixed croplands used for stubble grazing with some grassland and scrubland. Area 3 (A3) has a top grassy area mixed with shrubs and with few forested areas surrounded by tree crops with irrigation. Concerning Area 4 (A4), it is mainly covered by coniferous woodland with mixed crops on small patches (Figure 2). For the following analyses, 11 and 61 pixels corresponding to rainfed areas were selected for A3 LR and MR, respectively, to eliminate the effects of irrigation in the NDVI dynamics. The other areas are mainly conformed by rainfed crops, grasslands, and reforestation that do not present irrigation [43]. Therefore, all their pixels were used since no irrigation could affect their NDVI dynamics. The selection was made based on each pixel's average NDVI for summer months (June, July, and August). Pixels with a summer average below 30 and did not have peaks over 40 in its original time series, were selected to be analyzed, and these pixels match the grassy patch that crosses A3 from the top center to the right bottom corner.

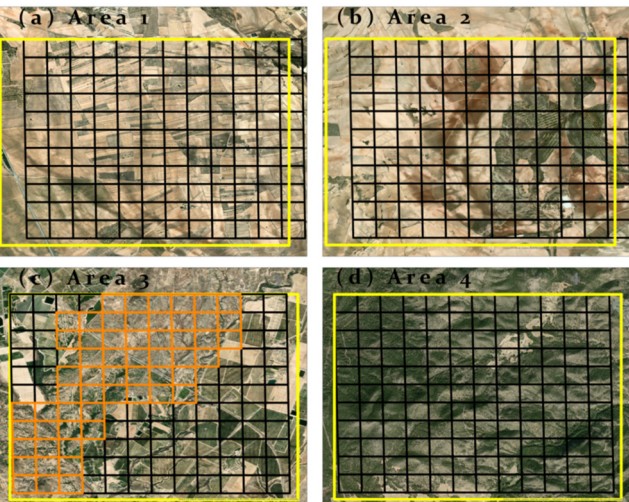

**Figure 2.** The four areas studied: (**a**) Area 1 with mostly stubble; (**b**) Area 2 with mixed crops and grasslands; (**c**) Area 3, with scrublands and mixed crops; (**d**) Area 4 with open woodland. In yellow, the area used for Low Resolution (LR); in black, the area used for Medium Resolution (MR) and orange, the selected pixels without irrigation. Source base map: Esri, DigitalGlobe, GeoEye, i-cubed, USDA FSA, USGS, AEX, Getmapping, Aerogrid, IGN, IGP, swisstopo, and the GIS User Community.

### 2.2. NDVI and Meteorological Data Collection

To study spatial resolution effects, the target areas were studied using $500 \times 500 \text{ m}^2$, 25 pixels for each area, and $250 \times 250 \text{ m}^2$, implying a set of 132 pixels (Figure 2). MOD09Q1.006 (LR) and MOD09A1.006 (MR) were collected from AppEEARS [44], downloading the RED (band 1) and NIR (band 2) values for the target areas at these spatial scales. Both had an 8-day temporal resolution from the beginning of 2002 through 2019, a total of 18 years of data. A maximum temporal resolution among the cited literature was found to be biweekly [16], and more commonly by month [15,17–21]. For each pixel series, R was used to calculate the Normalized Difference Vegetation Index (NDVI), using the following formula:

$$\text{NDVI} = 100 \times \frac{\text{NIR} - \text{RED}}{\text{NIR} + \text{RED}} \tag{1}$$

The possible NDVI values range from 0 to 100. The NDVI obtained values were then checked for quality. If the data were not categorized as ideal quality in the quality band from AppEEARS, less than 0.01% was deleted for every area. The gaps were filled using running averages with a gap interval of seven dates. The time series were then smoothed using the Savitzky–Golay method [45], with a window size of nine selected, based on the best-fitted outputs. The NDVI anomaly ($Z_{NDVI}$) was calculated by applying a z-score by date [46]:

$$Z_{NDVI} = \frac{\text{NDVI} - \mu_{NDVI}}{\sigma_{NDVI}} \tag{2}$$

where $\mu_{NDVI}$ is the average of all data in all available years that were measured in the same calendar date and $\sigma_{NDVI}$ is the corresponding standard deviation. In this new series, the seasonal variations of NDVI were eliminated. Daily meteorological data from the closest meteorological stations (Table A1) were also used [47]. Average temperature and accumulated precipitation were calculated for every eight days to match the NDVI dates.

To examine the variation of NDVI during the year, boxplots of NDVI and meteorological variables were used. Different phases were defined based on NDVI pattern during the year as this region presents harshness and aridity. These phases (*i*) were based on the trend of NDVI values: increasing, decreasing, or constant. A Chow test [48] of NDVI and time was used to confirm whether NDVI phases presented structural differences at the selected breaking points.

### 2.3. Correlations of NDVI with Meteorological Variables

The NDVI data were based on eight-day compound images. The image was selected based on criteria such as clouds and solar zenith. The temperature was the average for every eight days. Precipitation was accumulated every eight days, being coherent with the time scale of NDVI series. For exploring the existence of temporal patterns, we focused on two types of correlations:

Pearson's correlation coefficient of the NDVI values with Temperature (Temp) and Precipitation (Pp) at different phases ($i$) is:

$$\rho_{NDVI,Temp,i} = \frac{cov(\langle NDVI_i(t) \rangle, \langle Temp_i(t) \rangle)}{\sigma_{\langle NDVI_i(t) \rangle} \sigma_{\langle Temp_i(t) \rangle}} \tag{3}$$

$$\rho_{NDVI,Pp,i} = \frac{cov(\langle NDVI_i(t) \rangle, \langle Pp_i(t) \rangle)}{\sigma_{\langle NDVI_i(t) \rangle} \sigma_{\langle Pp_i(t) \rangle}} \tag{4}$$

where $\langle V_i(t) \rangle$ is the average of the 18 years of the variable $V$ ($NDVI$, $Temp$ or $Pp$) at time $t$ belonging to phase $i$; $\sigma_{\langle V_i(t) \rangle}$ is the standard deviation of the $\langle V_i(t) \rangle$.

Pearson's correlation coefficient of the NDVI series belonging to phase $i$, through the 18 years ($j$), with each of the climatic variables (*Temp* and *Pp*) at different time lags ($s$) is:

$$\rho_{NDVI,Temp,i}(s) = \frac{cov(NDVI_i(j,t), Temp(j, t-s))}{\sigma_{NDVI_i(j,t)} \sigma_{Temp(j,t-s)}} \tag{5}$$

$$\rho_{NDVI,Pp,i}(s) = \frac{cov(NDVI_i(j,t), Pp(j, t-s))}{\sigma_{NDVI_i(j,t)} \sigma_{Pp(j,t-s)}} \tag{6}$$

where $NDVI_i(j,t)$ are the $NDVI$ values at year $j$ and time $t$ that belong to phase $i$. The $Temp(j, t-s)$ are the temperature values at year $j$ and delayed $s$ times the lag time, which is eight days. Analogously, $Pp(j, t-s)$ can be defined.

### 2.4. Aridity Index and NDVI

The aridity index was calculated following [35], but instead of accumulating annually, we used the phases described by the NDVI patterns:

$$AI_i = \frac{P_i}{ETo_i} \tag{7}$$

where $P_i$ is the summation of the accumulated precipitation of each phase for each year and was analogously done for the accumulated potential evapotranspiration ($ETo_i$). Then, the average NDVI value for each phase was calculated. The cumulative aridity index and the cumulative average NDVI for each phase were plotted, and a linear regression was calculated to compare the four areas. A high slope would indicate an efficient use of its water resources.

## 3. Results

### 3.1. Interannual Variation

When yearly average NDVI, temperature, and precipitation were plotted at the two resolutions, different behaviors were found in these 18 years (Figures 3 and 4). In all the areas, NDVI at both resolutions were very similar. In A1 and A3, NDVI presents almost a constant value (A1 has 20 and NDVI nearly 30) as does temperature (15 °C in every area, except A3 where it reaches 18 °C). Precipitation is constant in A1 while in A3 it shows an increasing trend (A3 presents the lowest precipitation, mostly below 300 mm and the remaining areas range between 300 and 400 mm). On the other hand, A2 and A4 NDVI present a slight increase (with values around 20 for A2 and 40 for A4). In both areas, temperature and precipitation show different trends.

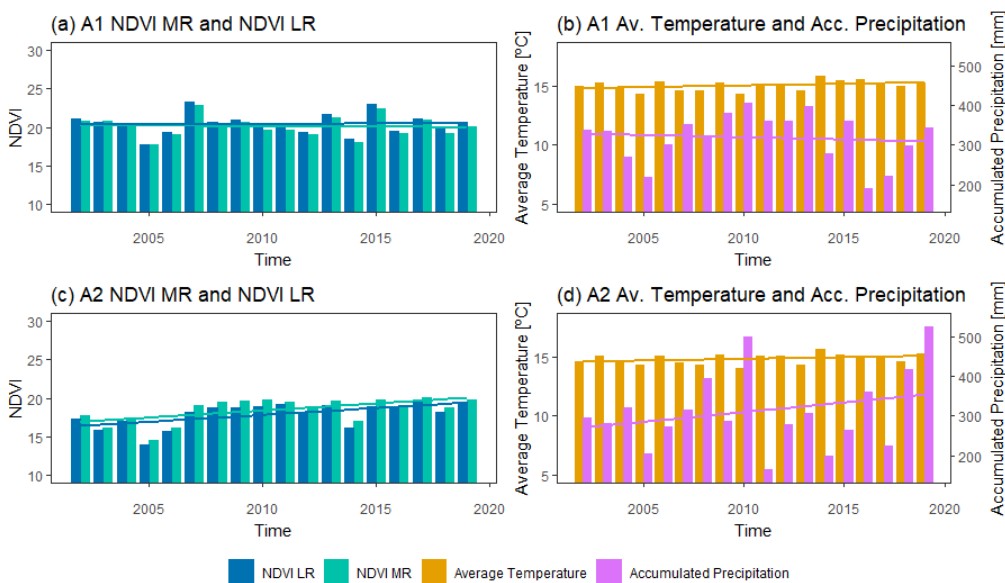

**Figure 3.** Bar plots of yearly average NDVI, temperature, and accumulated precipitation for A1 (**a,b**) and A2 (**c,d**) for Medium and Low resolutions.

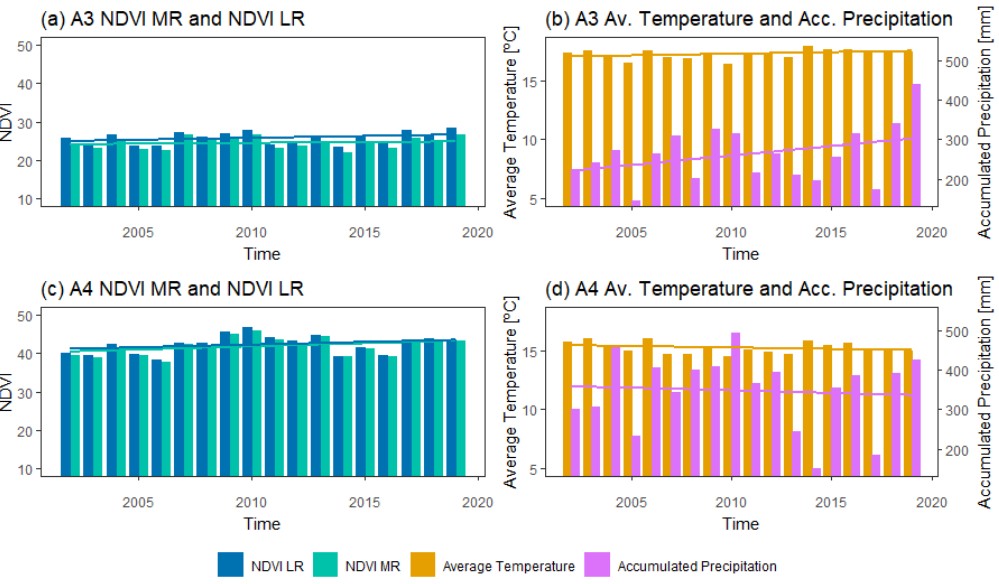

**Figure 4.** Bar plots of yearly average NDVI, temperature, and accumulated precipitation for A3 (**a,b**) and A4 (**c,d**) for Medium and Low resolutions.

To study whether these trends, visible in Figures 3 and 4, are statistically significant, a Mann–Kendall test [49,50] was applied in each area (Table 1). A1 shows a decreasing trend with NDVI MR and LR, but it is only significant with MR. On the other hand, A2 to A4 show an increasing trend, significant in both resolutions. The temperature increases in A1 to A3 and decreases in A4 and precipitation decreases in A1 and A4 and increases in A2 and A3. However, temperature and precipitation do not present significant trends, except for precipitation in A4, which decreases significantly (Table 1).

**Table 1.** Mann–Kendall test results of the four areas; * means that the trend was found significant. De: Decreasing; In: Increasing; Temp: Temperature; Precip: Precipitation.

| Areas | Trend | S-Statistics | Kendall's tau | *p*-Value |
|---|---|---|---|---|
| A1-MR | De * | −15,213 | −0.04 | <0.05 |
| A1-LR | De | −7466 | −0.02 | 0.34 |
| A1-Temp | In | 8849 | 0.03 | 0.03 |
| A1-Precip | De | −7547 | −0.02 | 0.34 |
| A2-MR | In * | 89,524 | 0.26 | <0.05 |
| A2-LR | In * | 98,292 | 0.28 | <0.05 |
| A2-Temp | In | 9115 | 0.02 | 0.25 |
| A2-Precip | De | −4599 | −0.01 | 0.55 |
| A3-MR | In * | 18,649 | 0.05 | <0.05 |
| A3-LR | In * | 42,338 | 0.12 | <0.05 |
| A3-Temp | In | 9912 | 0.02 | 0.21 |
| A3-Precip | De | −6017 | −0.02 | 0.44 |
| A4-MR | In * | 29,096 | 0.08 | <0.05 |
| A4-LR | In * | 24,762 | 0.07 | <0.05 |
| A4-Temp | De | −126 | −0.0004 | 0.98 |
| A4-Precip | De * | −17,068 | −0.05 | <0.05 |

### 3.2. Low and Medium Resolution NDVI Series

The average NDVI time series with LR and MR are shown in the left column of Figures 5 and 6 for the four areas studied. When the averages of all pixels for both resolutions are plotted, they tend to coincide. Nevertheless, we can differentiate them at the peaks and valleys of most years. In A1 and A2 (Figure 5), the two series are very similar. However, we can see some peaks where LR is higher for A1 and A2. This effect is more prominent in the A3 case. On the other hand, in A4, the differences are minor between the two resolutions. These peaks where LR is higher than MR are due to a few pixels and dates where the NDVI are much higher in both resolutions. However, because the MR has more pixels averaged, the smoothing effect is more noticeable. Since not all pixels were used for A3, this rise of LR is more conspicuous.

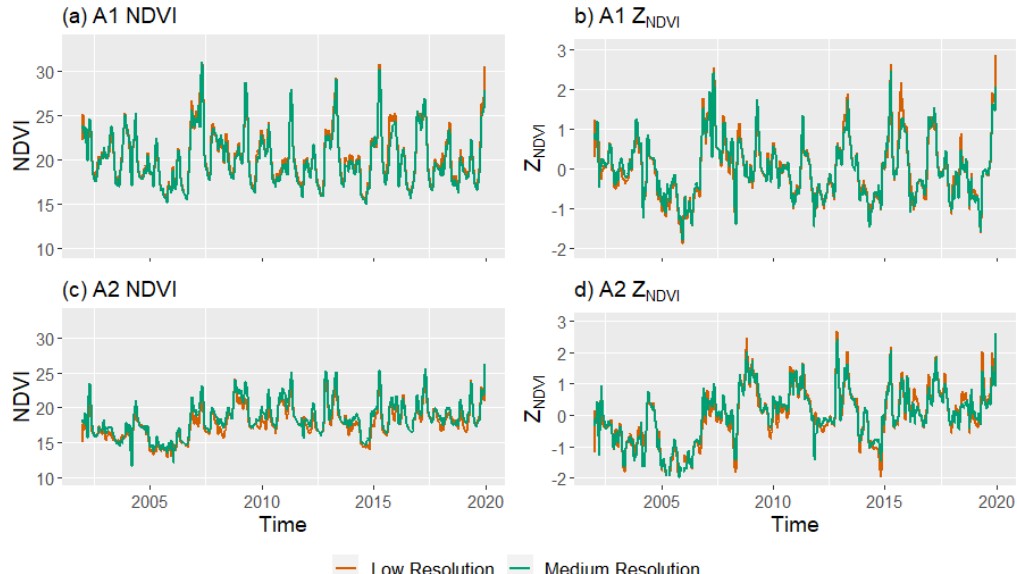

**Figure 5.** Series of average NDVI and $Z_{NDVI}$ for A1 (**a**,**b**) and A2 (**c**,**d**) for both resolutions: orange shows low resolution ($500 \times 500$ m$^2$) and green shows medium resolution ($250 \times 250$ m$^2$).

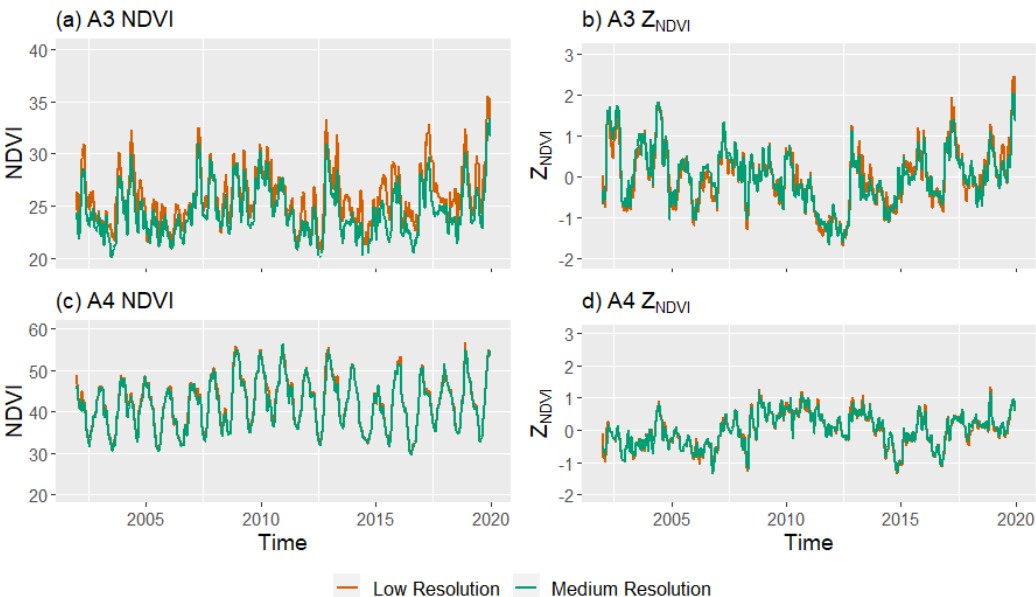

**Figure 6.** Series of average NDVI and $Z_{NDVI}$ for A3 (**a,b**) and A4 (**c,d**) for both resolutions: orange shows low resolution (500 × 500 m$^2$) and green shows medium resolution (250 × 250 m$^2$).

On the right column of Figures 5 and 6, we can observe the $Z_{NDVI}$. In all areas, there is a change in trend during 2006–2008. From 2000 until 2018, the year 2006 was one of the hottest, and after that, 2007 and 2008 were among the coolest years since 1996 [47], as can be observed in Figures 3 and 4. Looking at Figures 3 and 5, A1 and A2 show $Z_{NDVI}$ values in different years beyond +2 and −2. In these years, there are hotter temperatures for negative values with reduced precipitation. When $Z_{NDVI}$ goes above 2, high peaks on precipitation and temperature are lower than in other years. $Z_{NDVI}$ in A3 only goes above two once, at the end of 2019, coinciding with a strong peak in precipitation (Figures 4 and 6). Area 4 does not show values beyond these levels, although it shows how the trend rises or drops with these events. Regarding the resolution in A1 to A3, we can see peaks and valleys where LR has a higher frequency of extreme values. These differences are smoother for A4.

### 3.3. NDVI Patterns and Meteorological Variables

The LR and MR NDVI annual evolution and temperature and precipitation are shown in the following boxplots for each of the studied areas (Figures 7–10). The LR and MR NDVI values in A1 and A2 are very similar, then A3 presents an increase in these values, and A4 has much higher NDVI values than the rest. This pattern may be, at least in part, explained by increasing tree coverage in the areas. In each graph, the year was divided into different phases as the NDVI average trend changed.

A1 and A2 present a larger dispersion during the spring (March, April, and May). This dispersion is present but less prominent in A3, probably due to its less abundant precipitation. A4 exhibits a more consistent dispersion of values throughout the year with a reduction during the summer (June, July, and August), given by the almost complete tree coverage of this area. NDVI values decrease during the summer in all areas with a more visible trend in A4. When comparing NDVI and meteorological variables, the former shows greater heterogeneity among areas. NDVI and temperature appear to show a delay between their peak values. The average temperature peaks occur on 28 July in all the zones. The NDVI peaks on the 20 July in all the areas except A4, which showed it on 12 July (Figures 7 and 9 for A1 and A2, and Figures 8 and 10 for A3 and A4). Area A1 to A3 are more heavily influenced by agricultural practices, which may cause the delay between NDVI and temperature, which only takes eight days. Area A4, as open forest, does not have any irrigation regimes that can relate to an earlier peak of NDVI when temperatures rise. No different trends were found when boxplots were plotted for weeks, fortnight,

months, or season (data not reported). Meteorological trends remain similar regardless of the temporal scale.

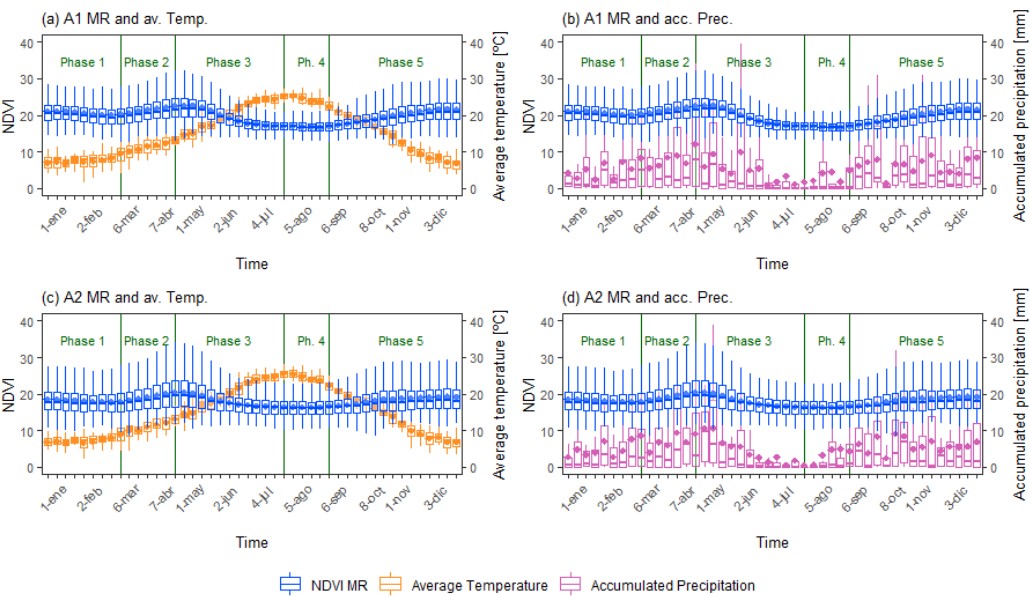

**Figure 7.** Boxplots of NDVI Medium Resolution (MR) for A1 (**a**,**b**) and A2 (**c**,**d**) in blue, average temperature in orange, and accumulated precipitation in purple.

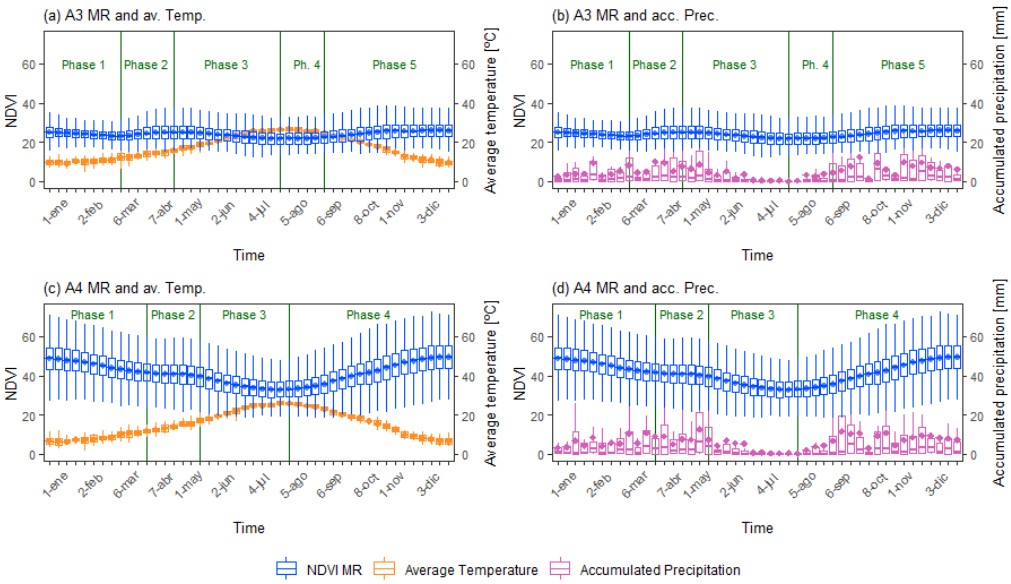

**Figure 8.** Boxplots of NDVI Medium Resolution (MR) for A3 (**a**,**b**) and A4 (**c**,**d**) in blue, average temperature in orange, and accumulated precipitation in purple.

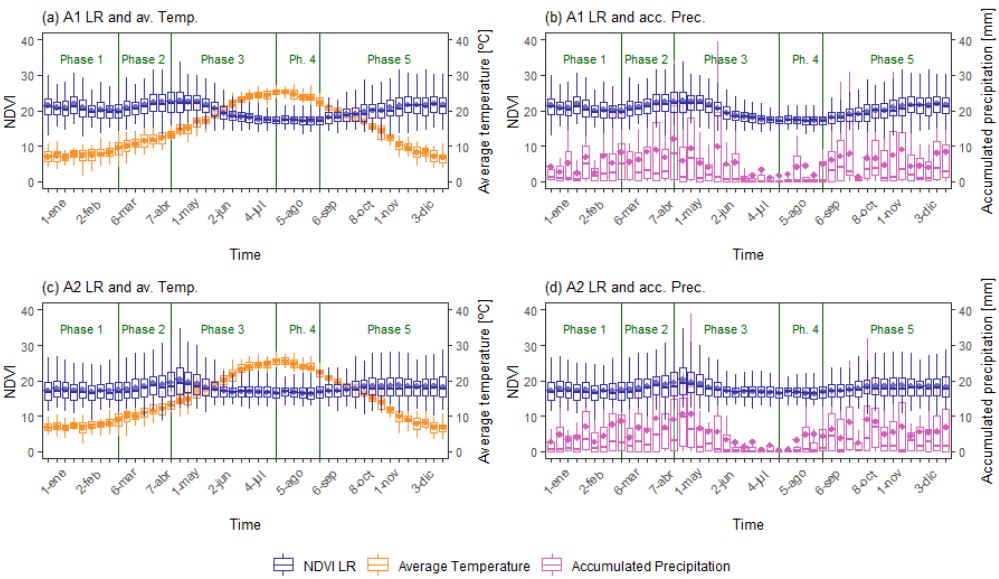

**Figure 9.** Boxplots of NDVI Low Resolution (LR) for A1 (**a**,**b**) and A2 (**c**,**d**) in dark blue, average temperature in orange, and accumulated precipitation in purple.

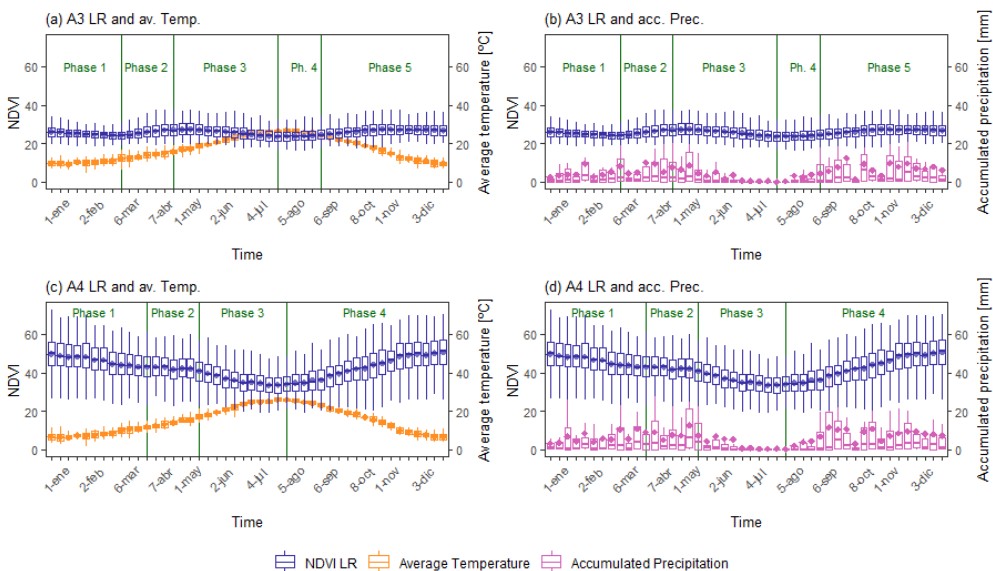

**Figure 10.** Boxplots of NDVI Low Resolution (LR) for A3 (**a**,**b**) and A4 (**c**,**d**) in dark blue, average temperature in orange, and accumulated precipitation in purple.

Because the NDVI trends did not match the change of climatic seasons, we separated the NDVI and meteorological variables according to the NDVI trend changes, as shown in Figures 7–10. Areas 1–3 were split into five phases, whereas A4 was split into four phases (Table 2).

**Table 2.** Division of phases following the behavior of NDVI. The selected phases in which there is an NDVI trend are shown in bold.

| Initial Dates | Area 1 to 3 | | Area 4 | |
|---|---|---|---|---|
| | **Phases** | **NDVI Trend** | **Phases** | **NDVI Trend** |
| 1-Jan | Phase 1 | Steady | **Phase 1** | **Decreasing** |
| 6-Mar | **Phase 2** | **Increasing** | | |
| 30-Mar | | | Phase 2 | Steady |
| 23-Apr | **Phase 3** | **Decreasing** | | |
| 17-May | | | **Phase 3** | **Decreasing** |
| 28-Jul | Phase 4 | Steady | **Phase 4** | **Increasing** |
| 6-Sep | **Phase 5** | **Increasing** | | |

### 3.4. Intra-Annual Regression by Phases

The regression analysis results between NDVI with temperature and precipitation by phases using the average of 18 years are shown in Figures A1–A4 (in Appendix B) for temperature and Figures A5–A8 (in Appendix B) for precipitation. Phases 1 and 4 were eliminated in A1–A3 and Phase 2 in A4, since their NDVI values were steady (Table 2). The regression analysis shows that for A1 to A3 (Figures A1–A3, Appendix B) temperature has a more substantial effect in NDVI in phase 2, 3, and 5, compared to phases 1 and 4, although phase 5 shows a slightly weaker relationship for A2, and especially A3. In A4, phases 3 and 4 show a strong relationship in the regression analysis and mild for phase 2 (Figure A4 in Appendix B).

The regression analysis for precipitation shows that for A1 to A3 (Figures A5–A7 in Appendix B), the herbaceous areas, NDVI in phase 3 is influenced by precipitation, while the others show a fragile relationship. No strong relationships are found in any phases in A4 (Figure A8). Due to the use of average values in the regression, differences between the two resolutions are small, although LR tends to have lower $R^2$ values than MR.

### 3.5. NDVI and Meteorological Series Correlation

The Chow test confirmed significant structural differences between phases for all areas (Table 3). Medium- and low-resolution NDVI produced similar results in the previous analyses. However, there were differences, particularly regarding the lagged responses. This paragraph describes the results of MR NDVI correlation analysis and discusses the differences with LR NDVI. The correlation values mentioned are the highest lagged correlation (Tables 4 and 5). NDVI has similar lags in each phase throughout A1 to A3. In phases 1 and 4, the NDVI values experience little change (with a difference within these phases between 4 and 10 in NDVI).

**Table 3.** Chow test of NDVI and time of all areas and phases, given with F-statistic and *p*-value for each continuous phase pair: phase 1 (P1), phase 2 (P2), phase 3(P3), and phase 4 (P4).

| Phases | A1 | | A2 | | A3 | | A4 | |
|---|---|---|---|---|---|---|---|---|
| | **F-Statistic** | ***p*-Value** | **F-Statistic** | ***p*-Value** | **F-Statistic** | ***p*-Value** | **F-Statistic** | ***p*-Value** |
| P1–P2 | 7.05 | 0.001 | 7.44 | 0.001 | 6.31 | 0.002 | 71.16 | 0.000 |
| P2–P3 | 4.67 | 0.010 | 3.33 | 0.037 | 4.54 | 0.011 | 108.00 | 0.000 |
| P3–P4 | 60.96 | 0.000 | 33.81 | 0.000 | 22.64 | 0.000 | 99.95 | 0.000 |
| P4–P5 | 53.44 | 0.000 | 34.65 | 0.000 | 48.39 | 0.000 | | |

**Table 4.** Lagged correlations of medium resolution NDVI and meteorological parameters (Meteo) for A1 and A2. Only the phases with an increasing or decreasing NDVI trend are included. For each phase, bold values show the strongest correlations among the different time lags tested.

| Area | Meteo | Phase | Lag Time (Days) | | | | | | |
|---|---|---|---|---|---|---|---|---|---|
| | | | 0 | 8 | 16 | 24 | 32 | 40 | 48 |
| A1 | Temp | Phase 2 | 0.23 | 0.3 | **0.32** | 0.29 | 0.29 | 0.19 | 0.13 |
| | | Phase 3 | −0.72 | −0.72 | −0.72 | −0.72 | **−0.73** | −0.7 | −0.66 |
| | | Phase 5 | −0.5 | −0.52 | −0.52 | −0.51 | **−0.52** | −0.51 | −0.5 |
| | Pp | Phase 2 | 0.11 | 0.19 | **0.26** | 0.24 | 0.28 | 0.28 | 0.2 |
| | | Phase 3 | 0.21 | **0.22** | 0.21 | 0.2 | 0.15 | 0.12 | 0.15 |
| | | Phase 5 | 0.08 | 0.08 | 0.11 | 0.13 | 0.17 | 0.23 | **0.25** |
| A2 | Temp | Phase 2 | 0.18 | 0.22 | 0.25 | 0.24 | **0.29** | 0.21 | 0.17 |
| | | Phase 3 | −0.58 | −0.58 | −0.59 | −0.59 | **−0.61** | −0.58 | −0.55 |
| | | Phase 5 | −0.35 | **−0.35** | −0.34 | −0.34 | −0.34 | −0.33 | −0.32 |
| | Pp | Phase 2 | 0.12 | 0.12 | 0.15 | **0.19** | 0.17 | 0.15 | 0.03 |
| | | Phase 3 | 0.17 | 0.19 | 0.22 | **0.23** | 0.19 | 0.17 | 0.17 |
| | | Phase 5 | 0.23 | 0.3 | **0.32** | 0.29 | 0.29 | 0.19 | 0.13 |

Average Temperature (Temp) and Accumulated Precipitation (Pp).

**Table 5.** Lagged correlations of medium resolution NDVI and meteorological parameters (Meteo) for A3 and A4. Only the phases with an increasing or decreasing NDVI trend are included. For each phase, bold values show the strongest correlations among the different time lags tested.

| Area | Meteo | Phase | Lag Time (Days) | | | | | | |
|---|---|---|---|---|---|---|---|---|---|
| | | | 0 | 8 | 16 | 24 | 32 | 40 | 48 |
| A3 | Temp | Phase 2 | −0.09 | 0.04 | 0.13 | 0.25 | **0.25** | 0.21 | 0.20 |
| | | Phase 3 | −0.04 | −0.06 | −0.09 | **−0.12** | −0.11 | −0.09 | −0.07 |
| | | Phase 5 | −0.02 | −0.03 | −0.04 | **−0.04** | −0.03 | −0.02 | −0.01 |
| | Pp | Phase 2 | **0.28** | 0.22 | 0.14 | 0.01 | 0.07 | 0.10 | 0.09 |
| | | Phase 3 | 0.06 | 0.08 | 0.15 | **0.19** | 0.16 | 0.18 | 0.17 |
| | | Phase 5 | **0.12** | 0.10 | 0.10 | 0.08 | 0.08 | 0.13 | 0.14 |
| A4 | Temp | Phase 1 | **−0.14** | −0.11 | −0.07 | −0.01 | −0.04 | 0.04 | 0.09 |
| | | Phase 3 | −0.71 | −0.72 | **−0.73** | −0.72 | −0.73 | −0.71 | −0.69 |
| | | Phase 4 | **−0.31** | −0.23 | −0.17 | −0.07 | −0.02 | 0.00 | 0.03 |
| | Pp | Phase 1 | 0.02 | 0.03 | **0.10** | 0.08 | 0.12 | 0.11 | 0.11 |
| | | Phase 3 | 0.25 | 0.24 | 0.25 | **0.25** | 0.24 | 0.21 | 0.23 |
| | | Phase 4 | 0.34 | **0.34** | 0.27 | −0.05 | −0.02 | 0.06 | 0.02 |

Average Temperature (Temp) and Accumulated Precipitation (Pp).

For this reason, temperature and precipitation have a minimal correlation with NDVI during these phases. Phase 2 shows a positive correlation of NDVI and temperature for these three areas: 0.32 for A1, 0.28 for A2, and 0.25 for A3. Phase 3 presents a moderate negative correlation for A1 and A2 (−0.73 and −0.61, respectively) and a low correlation (−0.12) in A3. Phase 5 has a low negative correlation for A1 (−0.32), A2 (−0.35), and A3 (−0.04). Area 4 always presents a negative correlation between temperature and NDVI, all lower than −0.4 except in phase 3, which has a stronger negative correlation of −0.73. Precipitation presents small positive values in all phases and areas, all below 0.35.

Lags on temperature for A1 to A3 in phase 1 and 4 range from 0 to 8 days, except for phase 4 in A2. All of these have very low values due to the constant behavior of NDVI values. In phase 2, the highest correlation between NDVI and temperature is found with a 16-day lag in A1, and 32-day lag in A2 and A3. Phase 3 has lags of 24 and 32 days for these three areas. Phase 5 differs more between areas, with a lag of 32 days for A1, eight days for A2, and 24 days in A3. Area 4 only shows a 16-day lag in phase 3, while the

rest of its phases present no lag. The differences between the areas may be related to two factors, differences between their vegetation and their hydrological regimes. Precipitation correlation lags show small similarities between all four areas. The scattered precipitation of these habitats and water scarcity makes it difficult to find a pattern across the areas. These disparities appear to be related to differences in heavy rains at specific times in each phase and area.

Comparing these results with the LR ones (Tables 6 and 7), correlation exhibited one of our analyses' significant differences between the two resolutions. The lag presented by the highest correlation with temperature was similar for MR and LR, except for some cases where there was an approximate difference of 8-days lag. However, phases 4 and 5 present more significant differences with a lag of 16 and 32 days, in some cases. The precipitation variable presents an almost constant smaller lag in the LR, ranging from 8 to 48 days. Partial correlation with NDVI, temperature, and precipitation was made, but no significant differences were found (Tables A2–A5, in Appendix C). They indicated that water availability is highly affected by temperature in this region.

**Table 6.** Lagged correlations of Low Resolution (LR) NDVI and meteorological parameters (Meteo) for A1 and A2. Only the phases with an increasing or decreasing NDVI trend are included. For each phase, bold values show the strongest correlations among the different time lags tested.

| Area | Meteo | Phase | Lag Time (Days) | | | | | | |
|------|-------|-------|------|------|------|------|------|------|------|
| | | | 0 | 8 | 16 | 24 | 32 | 40 | 48 |
| A1 | Temp | Phase 2 | 0.2 | 0.25 | **0.34** | 0.25 | 0.33 | 0.32 | 0.16 |
| | | Phase 3 | **−0.67** | −0.66 | −0.66 | −0.65 | −0.66 | −0.64 | −0.61 |
| | | Phase 5 | −0.41 | −0.43 | **−0.44** | −0.44 | −0.44 | −0.42 | −0.41 |
| | Pp | Phase 2 | **0.18** | 0.16 | 0.23 | 0.26 | 0.18 | 0.19 | 0.18 |
| | | Phase 3 | 0.19 | **0.19** | 0.19 | 0.21 | 0.15 | 0.13 | 0.18 |
| | | Phase 5 | **0.17** | 0.1 | 0.11 | 0.13 | 0.16 | 0.23 | 0.24 |
| A2 | Temp | Phase 2 | 0.16 | 0.16 | 0.23 | 0.2 | 0.3 | **0.33** | 0.22 |
| | | Phase 3 | −0.45 | −0.46 | −0.46 | −0.46 | **−0.48** | −0.46 | −0.46 |
| | | Phase 5 | −0.18 | **−0.2** | −0.19 | −0.19 | −0.17 | −0.16 | −0.15 |
| | Pp | Phase 2 | 0.19 | 0.06 | 0.16 | **0.2** | 0.15 | 0.03 | −0.03 |
| | | Phase 3 | **0.13** | 0.12 | 0.21 | 0.27 | 0.18 | 0.2 | 0.15 |
| | | Phase 5 | 0.2 | 0.25 | **0.34** | 0.25 | 0.33 | 0.32 | 0.16 |

Average Temperature (Temp) and Accumulated Precipitation (Pp).

**Table 7.** Lagged correlations of Low Resolution (LR) NDVI and meteorological parameters (Meteo) for A3 and A4. Only the phases with an increasing or decreasing NDVI trend are included. For each phase, bold values show the strongest correlations among the different time lags tested.

| Area | Meteo | Phase | Lag Time (Days) | | | | | | |
|------|-------|-------|------|------|------|------|------|------|------|
| | | | 0 | 8 | 16 | 24 | 32 | 40 | 48 |
| A3 | Temp | Phase 2 | 0.09 | 0.12 | 0.12 | **0.12** | 0.01 | 0.05 | 0.09 |
| | | Phase 3 | −0.40 | −0.41 | −0.41 | **−0.42** | 0.41 | −0.39 | −0.36 |
| | | Phase 5 | −0.28 | **−0.28** | −0.27 | −0.26 | −0.24 | −0.20 | −0.16 |
| | Pp | Phase 2 | **0.31** | 0.29 | 0.27 | 0.18 | 0.15 | 0.21 | 0.14 |
| | | Phase 3 | −0.08 | −0.02 | 0.04 | 0.09 | 0.08 | 0.12 | **0.15** |
| | | Phase 5 | 0.16 | 0.14 | 0.16 | 0.15 | 0.15 | **0.20** | 0.19 |
| A4 | Temp | Phase 1 | **−0.17** | −0.10 | 0 | 0.05 | −0.07 | 0.00 | 0.02 |
| | | Phase 3 | −0.66 | −0.65 | **−0.67** | −0.66 | −0.67 | −0.64 | −0.63 |
| | | Phase 4 | **−0.29** | −0.14 | −0.13 | −0.01 | 0.02 | 0.05 | 0.08 |
| | Pp | Phase 1 | **0.14** | −0.01 | 0.03 | 0.06 | 0.13 | 0.09 | 0.09 |
| | | Phase 3 | 0.25 | 0.25 | 0.22 | **0.26** | 0.22 | 0.18 | 0.22 |
| | | Phase 4 | **0.41** | 0.30 | 0.18 | −0.09 | −0.07 | 0.01 | −0.02 |

Average Temperature (Temp) and Accumulated Precipitation (Pp).

### 3.6. Aridity Index and NDVI

Figure 11 shows the four areas cluster into two groups. On one side, we find A1 and A2 intermingling. These two areas present lower cumulative NDVI than A3 and A4, and higher cumulative values of AI than A4 and A3. The AI for A4 was calculated based on its four phases, instead of five as was done for the other areas. However, this change does not affect the pattern and slope of this graphic. In the upper part of Figure 11, we find A3 and A4 showing efficient use of water resources. Their higher slope reflects the increase of NDVI as AI rises. With more typical Mediterranean vegetation, A3 and A4 have more efficient water use, particularly A3 with xerophilous shrubs as opposed to less xeric shrub vegetation, such as rosemary found in A4.

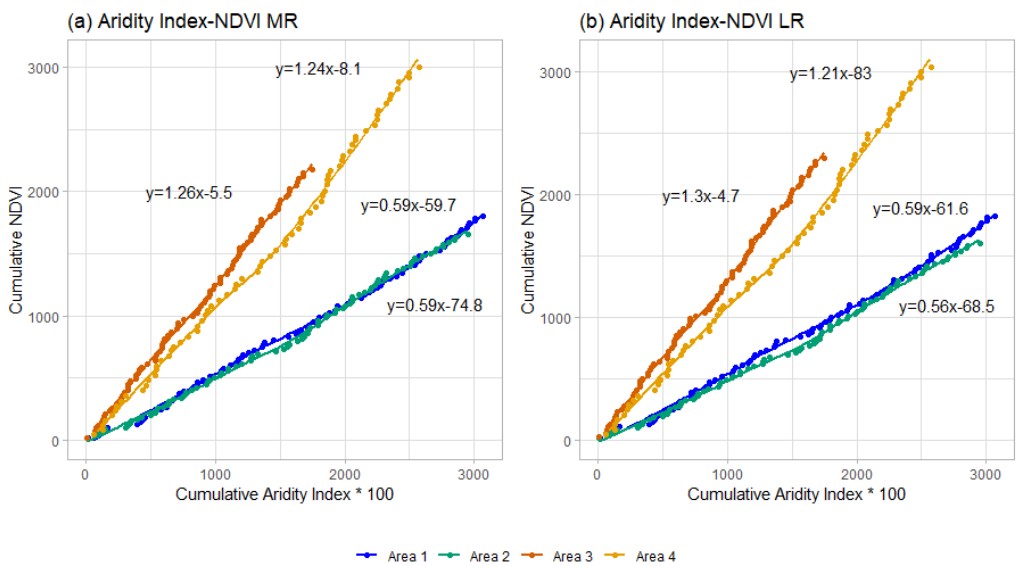

**Figure 11.** Linear regression analysis of cumulative AI and cumulative NDVI of all areas for MR (**a**), and LR (**b**). All $R^2$ values are above 0.99.

## 4. Discussion

### 4.1. NDVI Series Comparison

The average NDVI values for LR and MR of MODIS display relatively strong similarities. However, differences between them on their series' peaks and valleys can be detected (Figures 5 and 6). These differential patterns among areas suggest that it is caused by the difference of the averaged pixels within each area, showing that a finer scale is more representative of the area. On the other hand, some peaks and valleys are more pronounced, mainly when extreme meteorological events occur. These differences show a spatial difference among the pixels, which should be further researched.

These differences are confirmed when we compared the boxplots and the results of Pearson correlation and regression analysis. We found the significant differences in the boxplots among resolutions in A4, which is the most heterogeneous landscape, with a mix of tree-, scrub-, and grassy-dominated pixels. The medium resolution provided more significant results when studying their tendencies; the Mann–Kendall test for A1 with MR showed a tendency that was not visible with LR. Regression analysis provided more robust results for the relationship of NDVI and climatic variables with MR in A2 and A3, while A1 and A4 had smoother results than A2 and A3. Additionally, MR NDVI had more delayed lagged correlations. The use of medium or low resolution may depend on the spatial heterogeneity of the areas of study. This is probably due to a more detailed depiction of NDVI values in MR, as a smaller pixel size allows us to separate areas that could be spatially diverse. MR allows emerging a more lagged correlation without being tampered by the surroundings if an area dries at different times.

Using both resolutions shows a clearer image of the selected areas, especially highlighting those that are more heterogeneous. These results are in agreement with [31]. The use of $Z_{NDVI}$ and its comparison to the NDVI series also highlights the more significant changes during the series and the differences between the two resolutions. In particular, we find that extreme events are more pronounced when using LR as compared to MR.

*4.2. NDVI Data and Meteorological Variables*

NDVI values are highly related to water availability, as stated by [51]. NDVI responses are strongly linked to temperature in Mediterranean habitats, although this relationship weakens when precipitations are high [52]. In Murcia, the precipitations are scarce, with dry winters and almost no precipitation during the summer months. Furthermore, the temperature rises to its peak in July and August, leading to decreasing NDVI values. These values rise when the precipitation begins, and the temperature starts to descend. Our data show similar trends of higher NDVI values when precipitation increases, as highlighted by [53,54]. Area 4, located in the northwest county of Murcia, has relatively more water abundance and larger NDVI values.

Our correlation analysis shows that NDVI data are strongly and negatively correlated to average temperature when precipitation is strongly limited. The intraseasonal variation of the relationship of the NDVI and the meteorological variables has been approached by using different phases based on the NDVI behavior. The patterns of NDVI correlations with temperature change depending on the land use and the phase selected. Phases 1 and 4 show weak correlation values because the NDVI presents a steady pattern. On the other hand, phases 2, 3, and 5 have either increasing or decreasing NDVI trends. These phases present negative values except for phase 2 for A1 to A3 when the spring precipitation occurs. The correlations for NDVI and precipitation are very low, all below 0.35. However, the regression analysis highlights a stronger relationship for A1 to A3 for phase 3, when the precipitation starts high and steadily drops as temperature increases. A4, covered with trees, does not show a strong relationship between NDVI and precipitation where its precipitation in phase 2 is more limited than in the other areas. These changes among phases suggest that one of the critical factors affecting NDVI is water availability, matching other areas with semiarid and arid climates [55,56].

After characterizing the climatic variables and NDVI dynamics, we approached how the NDVI was affected by the AI, as a combination of precipitation and potential evapotranspiration, reflecting the temperature as well. This relation was established by accumulating NDVI and AI by the phases defined in each area. This allowed us to see how the different vegetation types representing each area respond differently to water availability. A3 and A4 show a more efficient response of NDVI to the rise of AI. Among these two areas, A3 exhibits a better response, in agreement with its more xeric vegetation, particularly in the selected top half pixels. No trees are found and xerophilous scrubs, such as esparto, inhabit the area. On the other hand, A1 and A2, representing different types of crops, show a smaller NDVI in response to a similar AI than A4, and higher than A3.

## 5. Conclusions

Comparison of the two scales of MODIS MOD09Q1.006 (MR) and MOD09A1.006 (LR) show that NDVI is scale dependent. These resolutions show differences, particularly when studying correlation and regression analysis. They are suggesting that medium resolution is more suited for spatial and lagged temporal patterns. However, when averaged, the trends are similar between these two resolutions. Lower resolution scales can be used when the studied areas are not spatially heterogeneous for temporal trend analysis, but larger resolutions scales are recommended on spatially diverse areas.

Among the climatic variables used, temperature shows the strongest relationship with average NDVI. Our results reveal that complex interactions of precipitation and temperature may explain real-time NDVI evolution. However, their behaviors vary across the phases. The use of phases based on NDVI patterns, instead of seasons, allows us to

describe a more realistic depiction of the arid environments, based on their vegetation dynamics. The study shows a strong positive correlation of NDVI with temperature when high precipitation occurs. Precipitation, however, shows a weak correlation with NDVI. The behavior of both climatic variables points to water availability as one of the major drivers of NDVI in Murcia, as it is suggested by the positive correlations of temperature and NDVI during phases where heavier precipitation occurs.

Aridity index and NDVI allowed us to cluster our four areas into two large groups, A1 and A2; mainly grazed wastelands showed a low increase of NDVI with a high aridity index. In contrast, A4 and A3 with a similar or lower aridity index presented a higher NDVI accumulation, showing more efficient use of available water; A3, especially, presented a higher slope than A4. However, more rangelands and other ecosystems should be analyzed to determine whether or not these differences can discriminate and characterize among other types of rangelands.

Intraseasonal relationship of NDVI with climatic variables was studied by splitting the analyses according to NDVI patterns. This allowed for viewing NDVI dynamics that were obscured when the seasonal division was followed. Intraseasonal and interseasonal characteristics should be taken into consideration in the definition of agrometeorological indexes in rangelands. This study provides a discriminating technique for rangeland management and policymakers. It is expected that future research will expand the knowledge of NDVI drivers at different scales, to develop tools and indexes that can help further comprehend vegetation communities of agricultural lands.

**Author Contributions:** Conceptualization, A.M.T., A.S.-R., C.H.D.-A. and E.S.; methodology, A.M.T., A.S.-R., E.I. and E.S.; formal analysis, A.M.T., A.S.-R., M.R.-R. and E.S.; writing—original draft preparation, E.S.; writing—review and editing, E.S., A.S.-R., C.H.D.-A., M.R.-R., A.R., E.I., P.E., B.S. and A.M.T., visualization, A.R., P.E., B.S. and E.S.; supervision, A.M.T.; funding acquisition, A.M.T. All authors have read and agreed to the published version of the manuscript.

**Funding:** Funding for this work was partially provided by Programme *Garantía Juvenil* from Comunidad de Madrid, and Boosting Agricultural Insurance based on Earth Observation data–BEACON project under agreement No. 821964, funded under H2020_EU, DT-SPACE-01-EO-2018-2020. The authors also acknowledge support from Project No. PGC2018-093854-B-I00 of the Spanish *Ministerio de Ciencia Innovación y Universidades* of Spain.

**Institutional Review Board Statement:** Not applicable.

**Informed Consent Statement:** Not applicable.

**Acknowledgments:** The data provided by Appears Team and AEMET are greatly appreciated.

**Conflicts of Interest:** The authors declare no conflict of interest. The funders had no role in the design of the study; in the collection, analyses, or interpretation of data; in the writing of the manuscript, or in the decision to publish the results.

## Appendix A

**Table A1.** UTM coordinates of each corner of the areas and the coordinate for their respective meteorological station.

| Areas | Area 1 | Area 2 | Area 3 | Area 4 |
|---|---|---|---|---|
| Xmin (m) | 660,520 | 635,796 | 650,064 | 614,504 |
| Xmax (m) | 663,020 | 638,296 | 652,564 | 617,004 |
| Ymin(m) | 4,284,420 | 4,273,396 | 4,225,985 | 4,219,731 |
| Ymax (m) | 4,286,920 | 4,275,896 | 4,228,485 | 4,222,231 |
| Stations | AL-10 (SIAM) | AL-06 (SIAR) | CI-32 (SIAR) | CR-32 (SIAM) |
| X station (m) | 675,585 | 630,946 | 652,564 | 615,466 |
| Y station (m) | 4,289,270 | 4,276,000 | 4,228,485 | 4,218,939 |

**Appendix B**

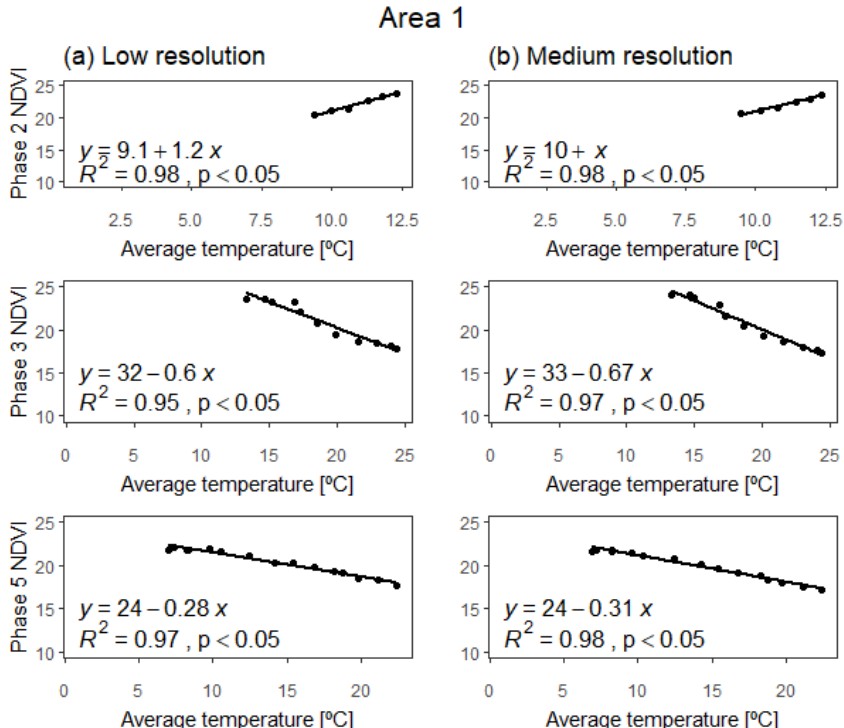

**Figure A1.** Regression analysis of average NDVI and temperature series separated by phases for LR (**a**) and MR (**b**) for A1.

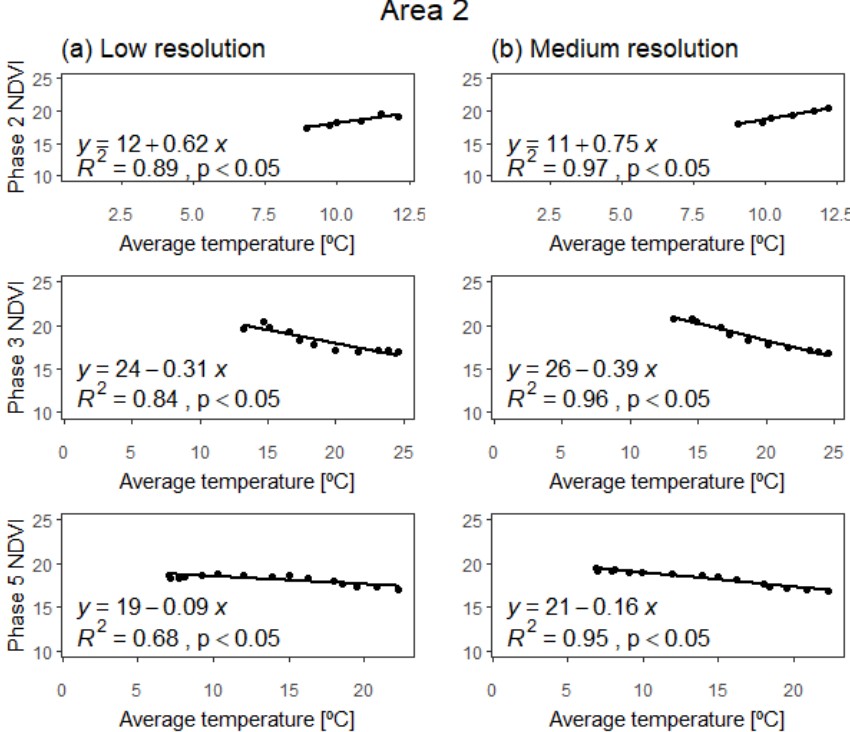

**Figure A2.** Regression analysis of average NDVI and temperature series separated by phases for LR (**a**) and MR (**b**) for A2.

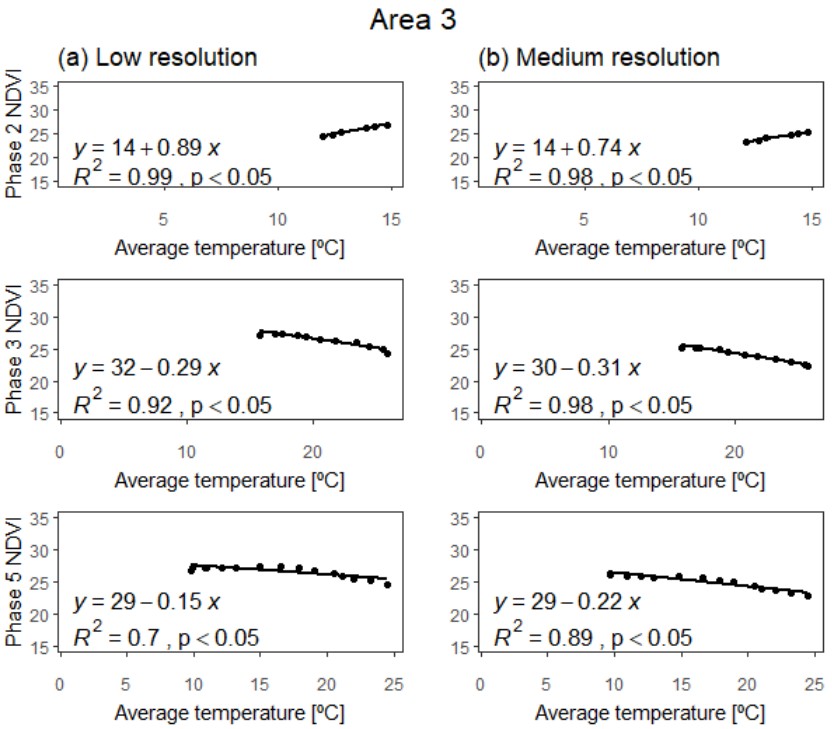

**Figure A3.** Regression analysis of average NDVI and temperature series separated by phases for LR (**a**) and MR (**b**) for A3.

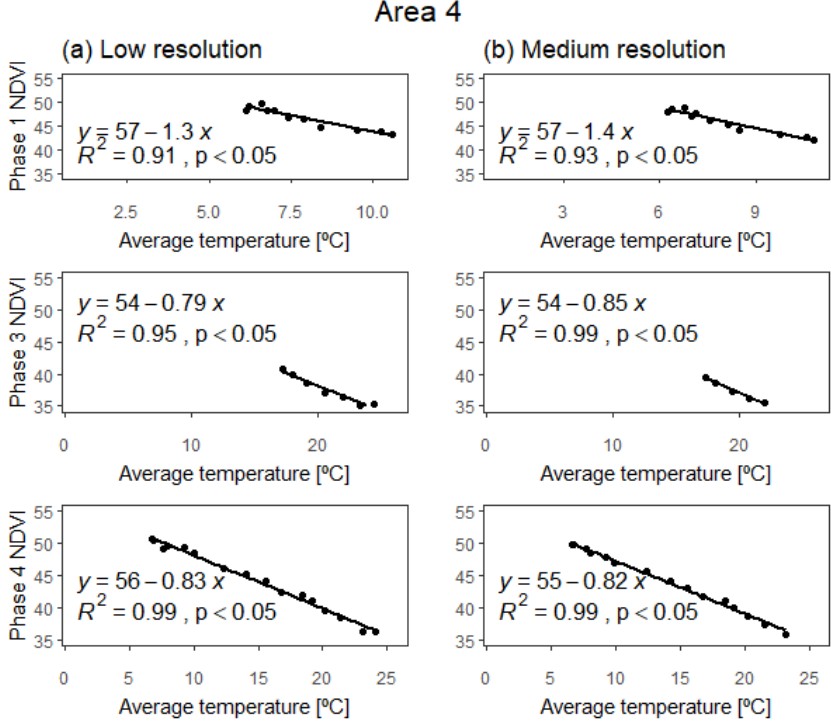

**Figure A4.** Regression analysis of average NDVI and temperature series separated by phases for LR (**a**) and MR (**b**) for A4.

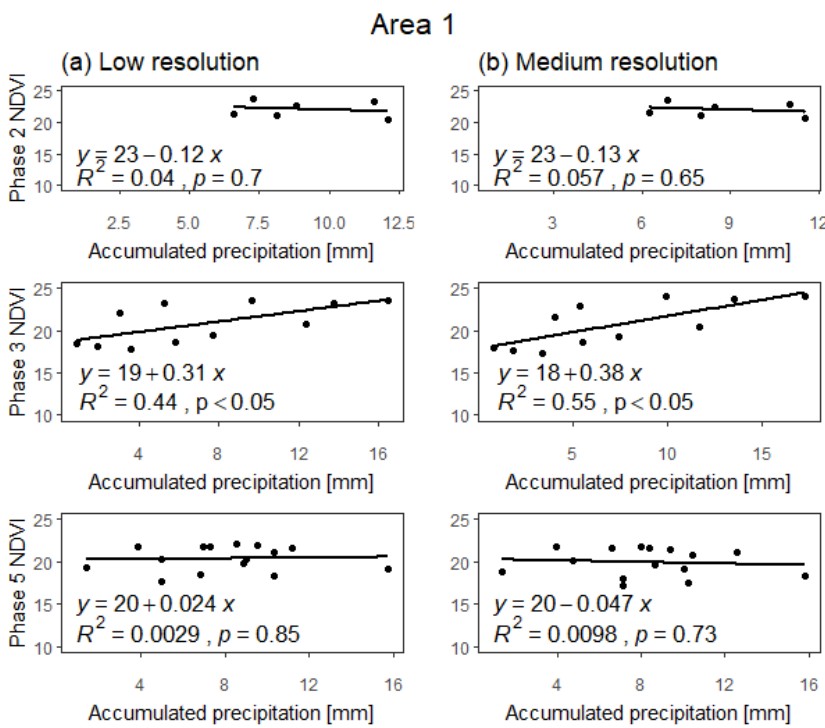

**Figure A5.** Regression analysis of average NDVI and precipitation series separated by phases for LR (**a**) and MR (**b**) for A1.

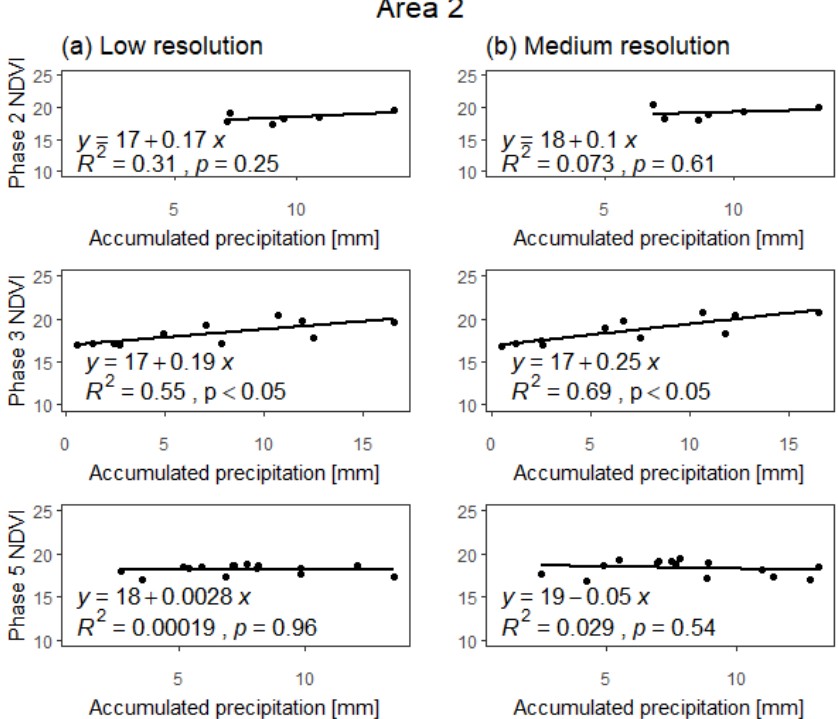

**Figure A6.** Regression analysis of average NDVI and precipitation series separated by phases for LR (**a**) and MR (**b**) for A2.

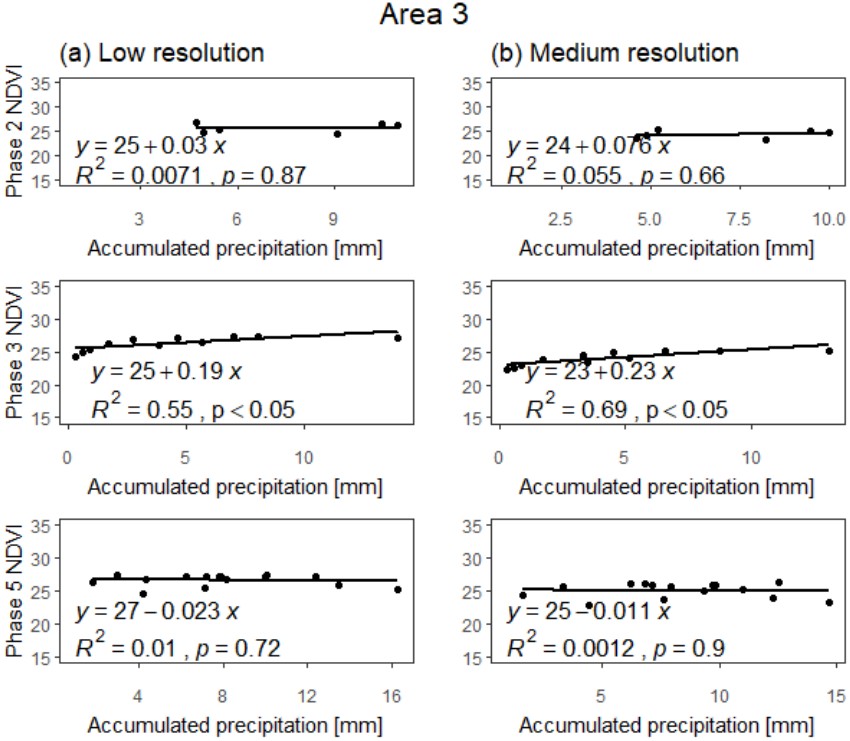

**Figure A7.** Regression analysis of average NDVI and precipitation series separated by phases for LR (**a**) and MR (**b**) for A3.

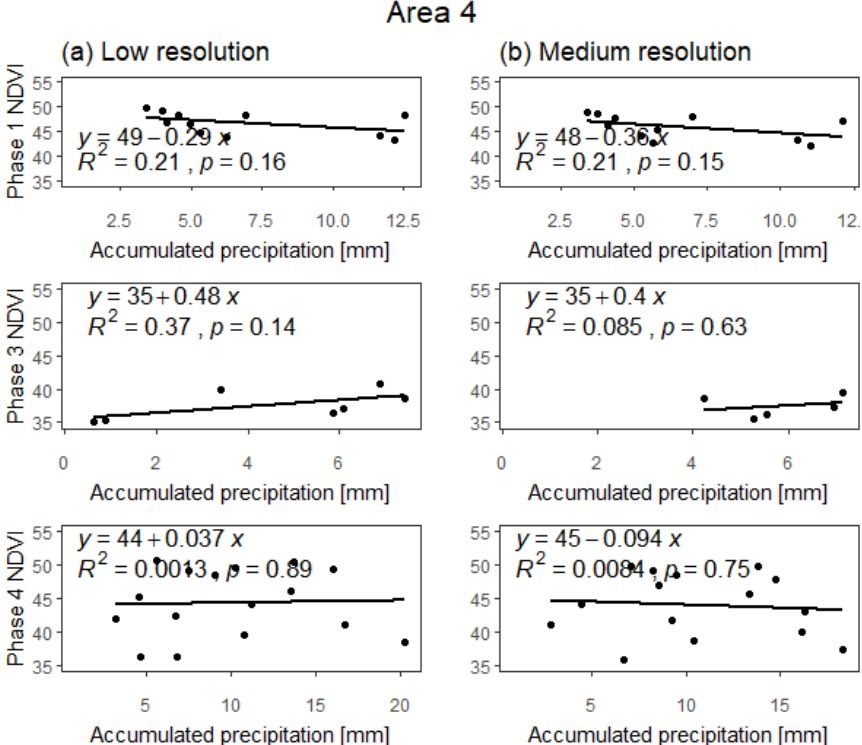

**Figure A8.** Regression analysis of average NDVI and precipitation series separated by phases for LR (**a**) and MR (**b**) for A4.

## Appendix C

**Table A2.** Lagged partial correlations of medium resolution NDVI and meteorological parameters (Meteo) for A1 and A2. Only the phases with an increasing or decreasing NDVI trend are included. For each phase, bold values show the strongest correlations among the different time lags tested.

| Area | Meteo | Phase | Lag Time (Days) | | | | | | |
|------|-------|-------|------|------|------|------|------|------|------|
| | | | 0 | 8 | 16 | 24 | 32 | 40 | 48 |
| A1 | Temp | Phase2 | 0.28 | 0.35 | **0.37** | 0.31 | 0.32 | 0.25 | 0.19 |
| | | Phase3 | −0.71 | −0.71 | −0.71 | −0.7 | **−0.72** | −0.69 | −0.65 |
| | | Phase5 | −0.5 | −0.51 | **−0.52** | −0.51 | −0.5 | −0.49 | −0.47 |
| | Pp | Phase2 | 0.19 | 0.27 | **0.32** | 0.27 | 0.31 | 0.32 | 0.23 |
| | | Phase3 | 0.13 | 0.17 | 0.16 | **0.19** | 0.01 | 0.08 | 0.08 |
| | | Phase5 | 0.08 | 0.07 | 0.09 | 0.08 | 0.11 | 0.15 | **0.16** |
| A2 | Temp | Phase2 | 0.21 | 0.24 | **0.27** | 0.24 | 0.3 | 0.25 | 0.18 |
| | | Phase3 | −0.57 | −0.57 | −0.56 | −0.57 | **−0.59** | −0.57 | −0.53 |
| | | Phase5 | −0.34 | **−0.35** | −0.34 | −0.34 | −0.32 | −0.31 | −0.29 |
| | Pp | Phase2 | 0.16 | 0.16 | 0.18 | 0.19 | 0.19 | **0.2** | 0.07 |
| | | Phase3 | −0.1 | −0.04 | 0 | 0.03 | 0.02 | **0.06** | 0.05 |
| | | Phase5 | 0.13 | 0.13 | 0.16 | 0.17 | **0.18** | 0.16 | 0.19 |

Average Temperature (Temp) and Accumulated Precipitation (Pp).

**Table A3.** Lagged partial correlations of medium resolution NDVI and meteorological parameters (Meteo) for A3 and A4. Only the phases with an increasing or decreasing NDVI trend are included. For each phase, bold values show the strongest correlations among the different time lags tested.

| Area | Meteo | Phase | Lag Time (Days) | | | | | | |
|------|-------|-------|------|------|------|------|------|------|------|
| | | | 0 | 8 | 16 | 24 | 32 | 40 | 48 |
| A3 | Temp | Phase2 | 0.11 | 0.18 | 0.24 | **0.32** | 0.32 | 0.28 | 0.22 |
| | | Phase3 | −0.48 | −0.48 | −0.49 | **−0.5** | −0.48 | −0.47 | −0.43 |
| | | Phase5 | −0.34 | **−0.34** | −0.33 | −0.32 | −0.3 | −0.28 | −0.24 |
| | Pp | Phase2 | **0.32** | 0.31 | 0.27 | 0.18 | 0.18 | 0.24 | 0.15 |
| | | Phase3 | −0.09 | −0.04 | 0.05 | 0.11 | 0.1 | 0.14 | **0.14** |
| | | Phase5 | 0.13 | 0.12 | 0.14 | 0.14 | **0.15** | 0.2 | 0.21 |
| A4 | Temp | Phase1 | **−0.14** | −0.11 | −0.06 | 0 | −0.05 | 0.02 | 0.07 |
| | | Phase3 | −0.69 | −0.7 | **−0.71** | −0.7 | −0.71 | −0.69 | −0.67 |
| | | Phase4 | **−0.23** | −0.15 | −0.17 | −0.07 | −0.03 | 0.03 | 0.04 |
| | Pp | Phase1 | −0.01 | 0.01 | 0.09 | 0.08 | **0.12** | 0.1 | 0.09 |
| | | Phase3 | −0.08 | −0.02 | 0 | 0.02 | 0.01 | 0.04 | **0.05** |
| | | Phase4 | 0.26 | **0.3** | 0.26 | −0.06 | −0.02 | 0.06 | 0.04 |

Average Temperature (Temp) and Accumulated Precipitation (Pp).

**Table A4.** Lagged partial correlations of low resolution NDVI and meteorological parameters (Meteo) for A1 and A2. Only the phases with an increasing or decreasing NDVI trend are included. For each phase, bold values show the strongest correlations among the different time lags tested.

| Area | Meteo | Phase | Lag Time (Days) | | | | | | |
|------|-------|-------|------|------|------|------|------|------|------|
| | | | 0 | 8 | 16 | 24 | 32 | 40 | 48 |
| A1 | Temp | Phase2 | 0.24 | 0.30 | 0.34 | 0.31 | **0.37** | 0.31 | 0.24 |
| | | Phase3 | **−0.07** | −0.03 | −0.01 | −0.04 | −0.01 | −0.05 | −0.04 |
| | | Phase5 | −0.45 | −0.46 | **−0.46** | −0.45 | −0.45 | −0.43 | −0.40 |
| | Pp | Phase2 | 0.23 | 0.27 | **0.30** | 0.24 | 0.27 | 0.32 | 0.24 |
| | | Phase3 | 0.12 | 0.18 | 0.23 | **0.28** | 0.06 | 0.11 | 0.10 |
| | | Phase5 | 0.11 | 0.09 | 0.11 | 0.10 | 0.12 | 0.17 | **0.18** |
| A2 | Temp | Phase2 | 0.20 | 0.20 | 0.26 | 0.25 | **0.33** | 0.29 | 0.24 |
| | | Phase3 | −0.06 | −0.01 | −0.12 | **−0.13** | −0.11 | −0.09 | −0.06 |
| | | Phase5 | −0.20 | **−0.21** | −0.20 | −0.19 | −0.17 | −0.15 | −0.13 |
| | Pp | Phase2 | 0.17 | 0.16 | 0.19 | **0.19** | 0.18 | 0.17 | 0.04 |
| | | Phase3 | 0.05 | 0.02 | 0.05 | 0.10 | 0.02 | **0.15** | 0.13 |
| | | Phase5 | 0.14 | 0.15 | 0.18 | 0.18 | **0.20** | 0.18 | 0.20 |

Average Temperature (Temp) and Accumulated Precipitation (Pp).

**Table A5.** Lagged partial correlations of low resolution NDVI and meteorological parameters (Meteo) for A3 and A4. Only the phases with an increasing or decreasing NDVI trend are included. For each phase, bold values show the strongest correlations among the different time lags tested.

| Area | Meteo | Phase | Lag Time (Days) | | | | | | |
|------|-------|-------|------|------|------|------|------|------|------|
| | | | 0 | 8 | 16 | 24 | 32 | 40 | 48 |
| A3 | Tmep | Phase2 | 0.11 | 0.18 | 0.25 | 0.34 | **0.36** | 0.30 | 0.23 |
| | | Phase3 | **−0.15** | −0.11 | −0.06 | −0.04 | −0.06 | −0.05 | −0.03 |
| | | Phase5 | −0.28 | **−0.28** | −0.27 | −0.26 | −0.24 | −0.20 | −0.17 |
| | Pp | Phase2 | **0.31** | 0.30 | 0.27 | 0.18 | 0.15 | 0.22 | 0.14 |
| | | Phase3 | 0.20 | **0.23** | 0.19 | −0.06 | 0.00 | −0.04 | −0.07 |
| | | Phase5 | 0.16 | 0.14 | **0.16** | 0.15 | 0.15 | 0.21 | 0.19 |
| A4 | Temp | Phase1 | **−0.14** | −0.08 | −0.01 | 0.03 | −0.05 | −0.01 | 0.04 |
| | | Phase3 | −0.68 | −0.69 | −0.70 | **−0.70** | −0.70 | −0.68 | −0.66 |
| | | Phase4 | **−0.20** | −0.14 | −0.15 | −0.05 | −0.02 | 0.02 | 0.03 |
| | Pp | Phase1 | 0.01 | 0.02 | 0.09 | 0.08 | **0.15** | 0.12 | 0.08 |
| | | Phase3 | −0.06 | 0.00 | 0.01 | 0.03 | 0.01 | 0.04 | **0.05** |
| | | Phase4 | 0.30 | **0.31** | 0.27 | −0.07 | −0.01 | 0.07 | 0.04 |

Average Temperature (Temp) and Accumulated Precipitation (Pp).

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
