# Peer review of "Normalized Difference Vegetation Index Temporal Responses to Temperature and Precipitation in Arid Rangelands"

_remotesensing, doi:10.3390/rs13050840_

Round 1
Reviewer 1 Report
The study employs multiple time-series remote sensing data to investigate the relationship between a well know vegetation index and the meteorological parameters of temperature and precipitation. The study is conducted in four sites located in the north-west Mediterranean region on the south part of Spain. The methods used are sound and well documented in the literature while the data, of two different spatial resolutions, have also been widely used for the calculation of the indices employed in this study (NDVI).
The main outcomes of the study is that there is a strong relationship between meteorological parameters and NDVI which is influenced by the physiological state of the vegetation so there is a strong intra-annual variation in this relationship. The second outcome is that the higher resolution data perform better in areas with higher spatial heterogeneity and the third outcome is that areas with less dynamic vegetation types (such as forests) have a more stable and higher NDVI.
My main concern about this study is the very low level of novelty. The relationship between vegetation status and NDVI has been widely studied and that’s why this particular index is widely used in several applications as the authors point out. The relationship between NDVI and meteorological parameters has also been widely studied using several different datasets. The study does not add anything new to the international remote sensing literature. Furthermore, it does not constitute a case study which could be useful at an operation level. It constitutes basic research with outcomes that are well known.
Another important limitation of this study is that three out of for study sites consists to a great degree of agricultural land which is probably irrigated. The irrigation regime is not considered in the study and since the results are obtained at a site level and not at pixel level the applied irrigation regime is highly likely to affect the calculated NDVI and subsequently its relationship with the meteorological parameters.
The study includes a huge amount of results which in most cases are discussed based on an optical inspection of the presented figures in order to draw the conclusions presented. I am not sure all these results are necessary in order to justify the main outcomes of the study.
Based on the above I am sorry to suggest rejection of the manuscript despite the huge amount of work done by the authors.
Reviewer 2 Report
This study analyzed correlation between NDVI and meteorological series. There are a number of issues the authors need to first address.
Temporal relationships between NDVI and key climatic variables including temperature and precipitation have been already examined in a number of studies (for instance, see a few links below).
https://link.springer.com/article/10.1007/s10666-011-9297-8
https://doi.org/10.1080/01431160210154812
https://doi.org/10.1016/j.ejrs.2020.08.003
https://doi.org/10.1080/01431160310001654428
https://doi.org/10.1080/01431160500306906
https://doi.org/10.1080/01431160110119416
https://doi.org/10.1080/01431160600567787
https://doi.org/10.1080/01431161.2013.787505
A more interesting study would comprise detailed analyses of the drivers of NDVI variation at various spatial and temporal scales.
The second way to make the research interesting would be to determine the amount of the total variance in NDVI which climatic variables (temperature, precipitation) can explain individually and jointly. Here, the authors would make use of multiple linear regression analysis based on adjusted R-squared.
The analyses were done purely in a conventional way and thus the originality of the work is limited. For instance, for correlation analyses using long-term series, the authors would examine the co-occurrences of sub-trends in NDVI, precipitation and temperature (see method via https://doi.org/10.2166/nh.2020.111 ). Besides, attributes of the temporal variation of NDVI could be examined through multiple hypothesis testing.
The authors determined significance of trends using parametric approach. The key assumption in the use of such a trend test is that the data is normally distributed. This assumption is mainly and so commonly violated by the way practical data series occur and eventually the trend results tend to be largely affected. Therefore, the authors should conduct trend test using
non-parametric or rank-based methods such as, Mann-Kendall test. The methodology for the test should be clearly presented in the paper.
Reviewer 3 Report
The authors investigated the response of NDVI to temporal dynamics of temperature and precipitation over four areas of an arid region in Spain. They analyzed the intra-seasonal variability with regression analysis and lagged correlation at two different spatial resolutions. The manuscript topic seems interesting but needs to be carefully revised and improved. I recommend the paper for publication after major and minor revisions described below are addressed.
- The title should be revised. At first glance the reader would wonder what NDVI means? You need to remove this abbreviation or write the long form instead.
- Line 20 – Add comma after “indices”.
- Line 76 – Concerning that correlation of NDVI with climatic variables can vary in different regions, please also mention the “studied region” of the literature that were discussed in the paragraph starting at line 76.
- In addition to human activities, the impacts of what other factors (e.g., evapotranspiration, land use change) on NDVI have been studied before? Such discussion should be added to the paragraph starting at line 76.
- Line 99 – Revise the table number.
- Line 101 – Revise figure 1 by removing the vertical line on the right hand side. Additionally, the text is too small (not readable) in this figure.
- Line 155 - replace “8” by “eight”.
- Line 214 – It has been shown that some peaks where LR is higher for A1 and, more prominently, for A2 than MR. How author justify higher peaks of LR for NDVI in figure 5 (left plot).
- Line 259 – Replace “Area 2 “ by “A2”
- There are too many figures and tables. Please bring some figures in the Appendix.
- Line 317 – Why strong correlation was not found in A3? What features of A3 can lead to such results?
- Line 422 – font type of tables 5 and 6 is not consistent with the rest of the manuscript.
Round 2
Reviewer 1 Report
The authors have significantly improved the manuscript and they have addressed my comments on the original version. I have no further comments and I wish to encourage the authors to continue their work.
Reviewer 2 Report
The authors made substantial improvement to the manuscript content.
Reviewer 3 Report
Fortunately, the authors have addressed all comments properly. My concern on discussing other factors impacting the NDVI dynamics have been addressed. Figures/tables have been also improved upon my request.
Overall, I recommend the manuscript for publication.